

# Morphological diversity of Western Balkan durum wheat landraces

Ana Velimirović[1], Sanja Mikić[2], Zoran Jovović[1], Dragan Mandić[3] and Novo Pržulj[4]

[1] Biotechnical Faculty, University of Montenegro, Podgorica, Montenegro
[2] Institute of Field and Vegetable Crops, Novi Sad, Serbia
[3] Agricultural Institute of Republika Srpska, Banjaluka, Bosnia and Herzegovina
[4] Academy of Sciences and Arts of the Republika Srpska, Banjaluka, Bosnia and Herzegovina

Corresponding author
Ana Velimirović,
ana.velimirovic@hotmail.com

## ABSTRACT

Wheat (*Triticum* spp.) is a vital food source for a substantial portion of the global population, with durum wheat (*Triticum turgidum* L. subsp. *durum* Desf) particularly significant in warmer regions like the Mediterranean. However, the aggressive introduction, spread and adoption of elite germplasm has led to crop genetic diversity loss, prompting efforts to preserve local durum wheat landraces. This study investigates the phenotypic diversity of 80 durum wheat landraces originating from the Western Balkan region, including accessions from Montenegro, Bosnia and Herzegovina, and Croatia. These landraces, locally known under traditional names such as 'Rogosija', 'Grbljanka', or 'Velja pšenica', represent a historical gene pool of durum wheat once widely cultivated until 1972 but subsequently abandoned. Historically, these landraces were valued for productivity, disease resistance, and resilience to drought and heat—traits well suited to Mediterranean conditions. Phenotypic traits were assessed across 17 morphological descriptors following International Union for the Protection of New Varieties of Plants (UPOV) guidelines. We observed wide trait variability, with high variation in ear and plant length, and limited variation in straw pith thickness and ear density. Strong correlations among certain traits suggest coordinated selection patterns or shared developmental pathways, while others may reflect distinct genetic or environmental influences. Our study identified 370 differentiated morphological types across 80 accessions, with most accessions displaying between four and six phenotypes. This demonstrates the extensive genetic variability within the collection. The normalized Shannon-Weaver index (H') across 17 traits averaged 0.59, indicating moderate to high diversity. Maximum H' values exceeded 0.80 for traits such as beak length, shoulder width of the lower glume, ear awn distribution, and recurved flag leaf frequency. Low variation in traits like straw pith thickness (H' = 0.05) and ear density (H' = 0.22) may indicate fixation or selection pressure. These findings provide valuable insights into Western Balkan durum wheat diversity, emphasizing the importance of considering both morphological traits and geographical origins in crop diversity studies. Overall, our study provides a foundation for future breeding efforts aimed at enhancing the agronomic performance and resilience of durum wheat cultivars.

## INTRODUCTION

Wheat (*Triticum* spp.) is a staple food and main food source for nearly 2.5 billion people (*Shiferaw et al., 2013*; *Tang et al., 2019*). Its adaptability to diverse climatic conditions, has made wheat the most widely cultivated cereal, grown on 220 million hectares worldwide, with an annual production of 770 million tons (*FAOSTAT, 2023*). While common wheat (*Triticum aestivum* L., 2n = 6x = 42, BBAADD) dominates global production accounting for about 95% of the total wheat growing area, the cultivation of durum wheat (*Triticum turgidum* L. subsp. *durum* Desf, 2n = 4x = 28, BBAA) has the greatest importance in warmer areas, including the Mediterranean (*Peng, Sun & Nevo, 2011*; *Boudiar et al., 2025*). The domestication of wheat, which began around 12,000 years ago during the "Neolithic Revolution", was marked by the cultivation of diploid einkorn (*Triticum monococcum* L.) in Anatolia and tetraploid emmer wheat (*Triticum turgidum* subsp. *dicoccum*) in the Levant (*Velimirović, Jovović & Pržulj, 2021*; *Heun et al., 1997*; *Peleg et al., 2011*). Over millennia, these ancestral forms diversified into locally adapted landraces. However, the 20th century Green Revolution profoundly altered global agriculture through the widespread adoption of high-yielding wheat varieties, intense use of mineral fertilizers, pesticides, and mechanization, tripling global wheat yields (*Evenson & Gollin, 2003*). This shift also accelerated the loss of genetic diversity, with locally adapted landraces disappearing at unprecedented rates (*Jovović & Kratovalieva, 2015*; *Qaim, 2020*). Evident and widespread decline of traditional crop diversity, commonly termed crop genetic erosion, has been documented for over a century and reported across regions and crops (*Khoury et al., 2022*; *FAO, 2025*). This erosion of crop genetic diversity also occurred in Montenegro and Western Balkan region, due to the aggressive introduction of elite germplasm that led to a noticeable erosion of the wheat gene pool (*Jovović, 2021*). Recognizing the danger of the disappearance of local durum wheat landraces, activities to collect and preserve durum wheat populations began in the 1950s. During that period, 125 autochthonous landraces of tetraploid wheat were collected. The relatively late establishment of the plant gene bank in Montenegro and the absence of a clear conservation program and financial resources contributed to permanent loss of 36% of the collection (*Jovović et al., 2017*; *Velimirović et al., 2023*). These landraces, once widely cultivated for their resilience to heat, drought, and disease, hold immense genetic potential for addressing contemporary agricultural challenges (*Xue et al., 2012*; *Tan et al., 2019*; *Gessese et al., 2019*; *Sahu et al., 2022*; *Velimirović et al., 2023*). Their anticipated capacities for accommodating current needs of the mankind to tackle climate challenges stemmed from their heterogeneous genetic makeup. The local wheat landraces consist of a mixture of diverse homozygous lines forming a wide genetic base that provides adaptiveness to variable adverse environmental conditions. The substantial number of local durum wheat accessions preserved in gene banks is often underexploited due to limited phenotypic evaluation, which hampers their integration into breeding programs (*Pignone et al., 2015*). Morphological traits defined by the International Union for the Protection of New Varieties of Plants (UPOV) provide a robust foundation for the initial characterization of accessions. Their application enables the identification of distinct phenotypes and deepens the understanding of intraspecific

diversity, which underpins the effective use of genetic resources. These primary phenotypic insights support broader breeding efforts aimed at improving traits such as adaptability and resilience, water productivity, lodging resistance, grain yield and phenology (*Foulkes et al., 2011*; *Reynolds et al., 2011*; *Sánchez et al., 2023*).

Although local genetic resources are recognized as valuable reservoirs of desirable genes for crop improvement, the extent and pattern of genetic variation in tetraploid wheat landraces from the Western Balkans, maintained in the Montenegrin gene bank, remain insufficiently studied. The first tetraploid free-threshing wheats (*Triticum durum* Desf. and *Triticum turgidum* L.) most likely arrived in Montenegro *via* maritime routes from Greece or southern Italy through multiple introductions, which contributed to increasing their genetic diversity. Additionally, some new varieties and forms may have arisen locally through prolonged evolutionary processes. Their cultivation was primarily associated with the Adriatic climate, occurring in the coastal zone and river valleys up to 600 m above sea level. Until the early 1970s, these species dominated wheat production in southern Montenegro, after which tetraploid free-threshing wheats declined rapidly (*Jovović et al., 2017*). Local populations of T. *turgidum* and related tetraploid wheats are traditionally known as Rogosija, Velja or Velika, names reflecting their larger plants, ears, grains, and stronger straw compared to other wheat species (*Pavićević, 1975*). Although a conservation program was initiated in the mid-20th century, resulting in the collection of over 150 diploid and tetraploid wheat populations in Montenegro and Herzegovina between 1955 and 1964, these accessions have yet to undergo detailed characterization, with available data remaining scarce (*Jovović et al., 2012*).

Morphological markers remain an essential tool for assessing genetic diversity in crop species due to their high heritability and practicality (*Al-Ashkar et al., 2020*; *Haque et al., 2021*).

Standardized protocols established by International Plant Genetic Resources Institute (IPGRI) and UPOV have facilitated the evaluation of phenotypic traits in wheat diversity studies (*Rabieyan et al., 2023*; *Fiore et al., 2022*). Besides morphological characteristics, certain phenological and agronomic traits (such as days to heading and plant height) are also used as descriptors. These traits exhibit broad-sense heritability values greater than 0.80, implying strong genetic control and limited environmental influence (*Gharib et al., 2021*).

The objective of this study is to investigate whether the Montenegrin gene bank collection of durum wheat landraces contains significant phenotypic diversity that remains undercharacterized and to evaluate whether this diversity can be effectively assessed using standardized morphological and agronomic descriptors. Accordingly, this research aims to assess the extent of the phenotypic diversity and analyze the efficiency of available descriptors—defined as their capacity to discriminate among landrace phenotypes—for the classification of Western Balkan durum wheat landraces. Additionally, it seeks to provide insights into adaptive traits exhibiting broad variability relevant for conservation and breeding.

## MATERIALS & METHODS

### Plant material and field experiment

The study included 80 durum wheat accessions collected from traditional farming systems across the Western Balkan, representing a spectrum of local agro-ecological zones. The geographic coordinates, altitudes, and Köppen–Geiger climate classifications of the collection sites were determined using 1-km resolution climate classification maps (*Beck et al., 2018*). Detailed information on each collection site is provided in Table S1. These accessions are part of the national gene bank collection and are listed with their corresponding phenotype codes in the same table. The field experiment was carried out in the 2020–2021 season at the Research Unit in Banjaluka (Bosnia and Herzegovina). The experimental site is characterized by alluvial soils formed on river sediments, with a loamy texture and balanced proportions of sand, silt, and clay, supporting good water retention and aeration capacity (*Tvica & Tunguz, 2023*). The growing season was marked by moderate temperatures and variable precipitation, with average monthly temperatures ranging from 1.5 °C in January to 22.3 °C in July, and total rainfall highest in December and July (Table S2). Standard agronomic practices for wheat cultivation were followed, including soil preparation and the application of NPK fertilizer (8:24:24) at 300 kg ha$^{-1}$. Considering the heterogeneous genetic background of the local durum landraces and to capture diversity within the population, each of 80 landraces was represented with 60 ears. Seeds of each accession were sown by hand at a depth of five cm in a plot consisting of twenty 1 m-long rows. Each of 20 rows was sown using the seeds from a single ear of the landrace. Rows were spaced 0.2 m apart, with 1 m between plots to minimize inter-plot interference and avoid unwanted pollen movements, thus ensuring seed purity. During the growing season, three individuals from each row were randomly selected for measuring to account for genetic variability totaling 60 samples per each landrace. Due to the exploratory nature of this study and the limited availability of seeds, no replicated block design or standard checks were included. This approach aligns with standard gene bank operations during seed renewal processes. Preliminary characterization using morphological descriptors is recommended, particularly for traits with high heritability, to be conducted alongside seed regeneration (*FAO, 2014*). Although analyses based on a single season and environment may limit the robustness of statistical inferences, there are scientific studies that have successfully employed UPOV descriptors, morphological and phenological data to characterize wheat during a single growing season (*Eticha et al., 2005*; *Zarkti et al., 2012*; *Chauhan et al., 2020*).

### Morphological characterization

Morphological traits were assessed following the guidelines for distinctness, uniformity and stability (DUS) established by UPOV. Nineteen morphological descriptors were used, encompassing traits related to the coleoptile, flag leaf, ear, and glumes, among others (Table S3). Data were scored by harvesting a random sample of three representative plants from all 20 rows in the plot (60 plants in total), which represents a landrace. To further refine the characterization, phenotypic traits were grouped based on their relevance to specific plant organs (*e.g.*, ear, flag leaf, plant habit) in accordance with UPOV standards

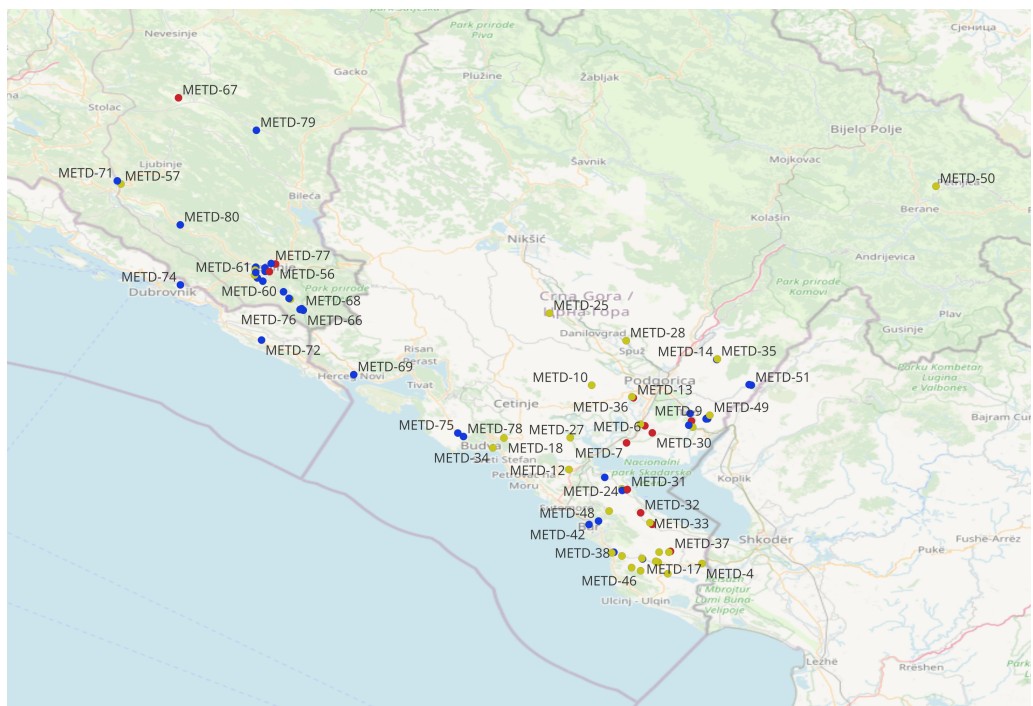

**Figure 1** **Geographic location of the 80 collection sites of durum accessions collected in the Western Balkan Peninsula (geographically proximate sites may overlap).** ©OpenStreetMap contributors (ODbL).

(Table S4). Data were summarized and analyzed collectively to better assess diversity patterns within and among accessions.

## Data analysis

Phenotypes with all identical scores across all traits (duplicates) were removed prior to the analyses. Phenotypes with at least one trait score different from the others are considered unique. For the grouping of genotypes based on morphological data, agglomerative hierarchical cluster analysis (AHC) was applied using Ward's method in XLSTAT (Addinsoft, Seattle, WA, USA; https://www.xlstat.com/). The categorical data obtained from morphological descriptors were used to calculate the normalized Shannon diversity index (H') for measuring of morphological diversity (*Shannon & Weaver, 1949*). Multiple correspondence analysis (MCA) was used to visualize obtained categorical variables and explore relations among morphological traits in R version 4.4.1 (*R Core Team, 2024*), packages FactoMineR v.2.7 (*Lê, Josse & Husson, 2008*) and Homals v. 1.0-10 (*De Leeuw & Mair, 2009*). The geographic coordinates of all accession collection sites were recorded and mapped using QGIS 3.34 (Fig. 1) (*Durum Wheat Accessions Sampling Sites Location: QGIS Development Team, 2024*). The analyzed landraces originated from 80 distinct collection sites, with their co-ordinates and corresponding toponyms listed in Table S1.
Trait frequencies were calculated by dividing the number of accessions exhibiting a specific category (or grade) of a morphological characteristic—based on UPOV descriptors—by the total number of accessions analyzed.

For each trait the normalized Shannon-Weaver diversity index (H') was calculated as

$$H' = \frac{-\sum_{i=1}^{n} p_i \log_2(p_i)}{\log_2(n)}$$

where $p_i$ is the relative frequency of the $i$-th category, and $n$ is the total number of categories for the trait. H' ranges between 0 (no diversity, only one category observed) and 1 (maximum diversity, all categories equally represented). The H' values were also averaged for specific plant organs grouped according to UPOV guidelines (*International Union for the Protection of New Varieties of Plants, 2012*).

## RESULTS

### Cluster analysis of obtained morphological data

Morphological characterization of 80 durum wheat accessions revealed significant phenotypic diversity, with 370 distinct morphological types (phenotypes) identified (Table S1). The number of phenotypes per accession ranged from 1 to 11, with most accessions exhibiting four to six phenotypes. The morphological differentiation of 370 unique phenotypes reveals the heterogeneous genetic backgrounds of local durum landraces as an adaptive strategy for specific agro-ecological microenvironments.

Measured morphological traits grouped all phenotypes into three AHC classes of 180, 135 and 57 genotypes, respectively (Fig. 2). All 180 phenotypes from the Cluster 1 (denoted as blue circles) did not exhibit a geographic clustering pattern, since they were collected from dispersed locations, across the northern parts of the Dalmatian coast, Herzegovina, the Montenegrin coast and near Skadar Lake and, one site in the northern mountainous region of Montenegro. Most of the collection sites from class two (133 out of 135 phenotypes denoted as yellow circles) and three (54 out of 57 phenotypes denoted as red circles) distributed around Skadar Lake, and north from the Skadar Lake, in the vicinity of the capital city Podgorica.

### Diversity of morphological traits

The normalized Shannon-Weaver index (H'), used as an indicator of morphological diversity of 370 phenotypes, ranged between 0 and 0.92 (Fig. 3). All accessions belonged to the alternative seasonal type without hairiness on external surface of lower glume, indicating fixed traits (Table 1 and Fig. 3). Low diversity values were measured for lower glume: hairiness of external surface of the lower glume (H' = 0.03) with only one accession with thin cross section, while all others had medium thickness. The majority of accessions were characterized by lax ear density (96%), while only 4% had medium ear density, resulting in low H' values (0.23).

The length of awns at the tip relative to the length of ear lacked diversity, since 91% of the phenotypes had the grade "equal" for this trait (H' = 0.30). The normalized diversity index between 0.60 and 0.70 was determined for ear length, ear glaucosity, and glaucosity

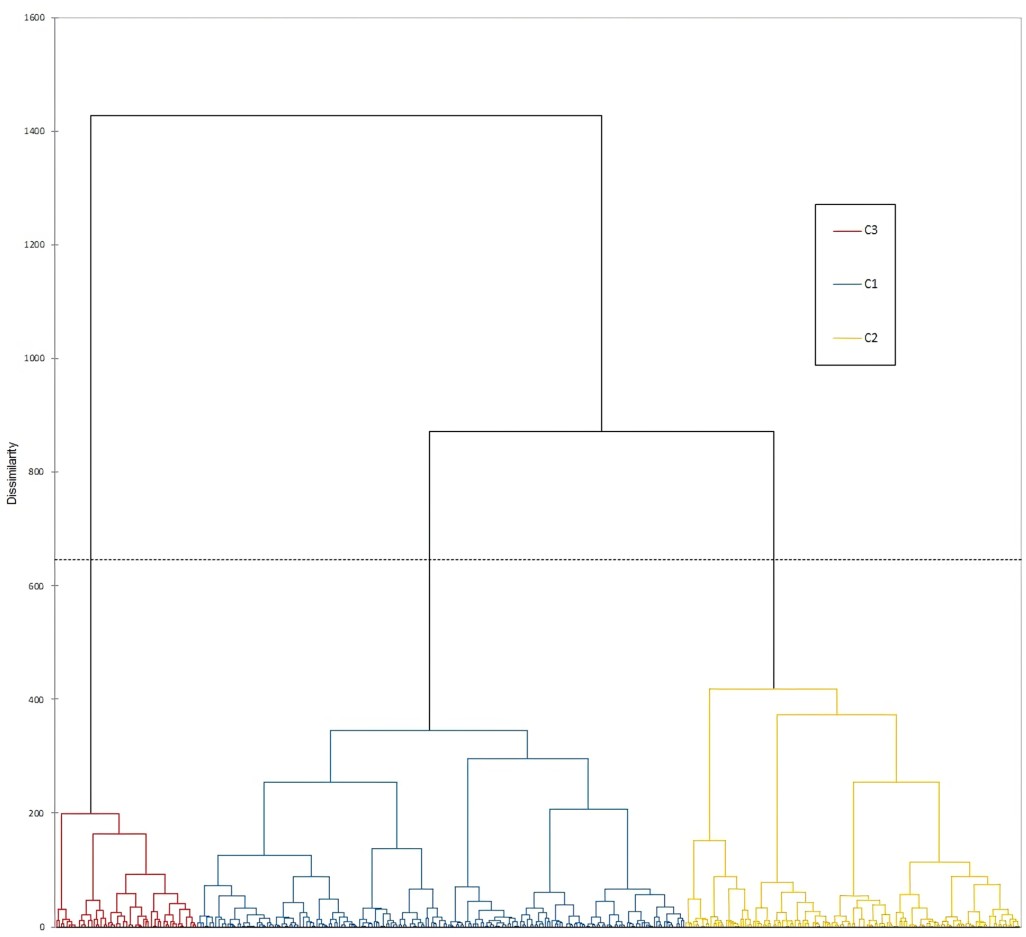

**Figure 2** **Agglomerative hierarchical cluster analysis of 370 durum wheat phenotypes.** The accessions included in classes one, two and three according to AHC are presented in blue, yellow and red circles, respectively.

on the lower side of flag leaf blade and curvature of the beak of the lower glume. Values between 0.70 and 0.80 were measured for plant length and growth habit, anthocyanin coloration of flag leaf auricles, time of ear emergence, and shape of the shoulder of lower glume. Finally, an even distribution of all grades and a Shannon-Weaver index above 0.80 were measured for four traits: length of beak and width of shoulder of lower glume, anthocyanin coloration of coleoptile, distribution of ear awns and frequency of plants with recurved flag leaves. Among trait groups, flag leaf and lower glume traits recorded the highest diversity (H' = 0.74 and 0.66, respectively), whereas ear and plant traits showed lower values (H' = 0.57 and 0.50, respectively).

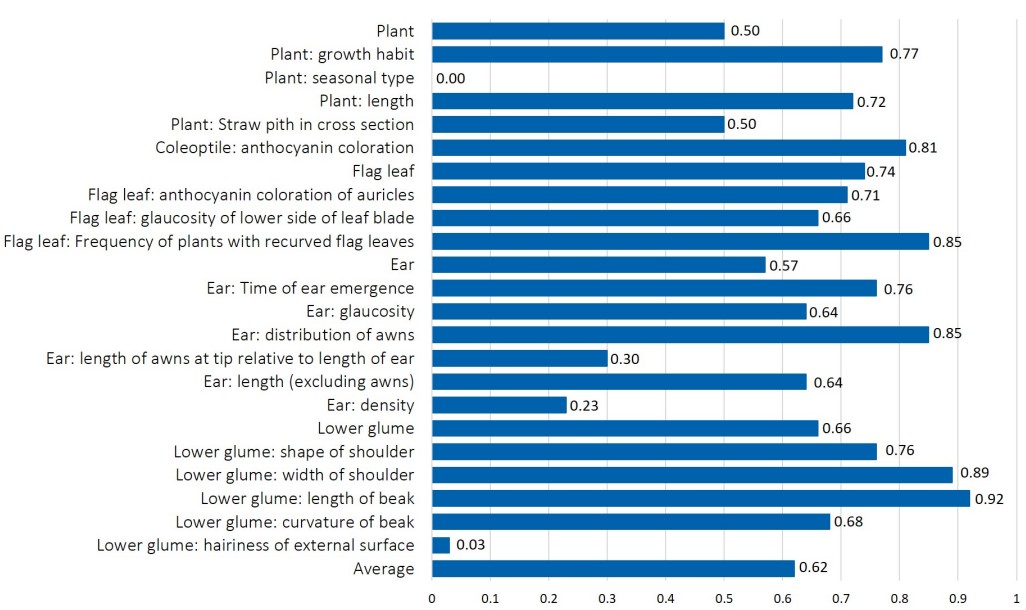

**Figure 3** The normalized Shannon-Weaver diversity index (H') calculated for mean value for each of 19 morphological characteristics, and for the mean values for plant, flag leaf, ear, and lower glume trait groups in durum wheat collection.

## Grouping of phenotypes and discriminative power of morphological traits

The multiple correspondence analysis (MCA) revealed limited clustering within the Western Balkan durum wheat collection, with dimensions 1 and 2 explaining only 1.8% of the total variability (Fig. 4).

Phenotypes were closely grouped, except for one outlier, METD-52/01, a phenotype from the Rogosija landrace characterized by white ears, dark awns, and very short stature (group 1, Tables S1 and S3). Excluding this outlier and the descriptor for anthocyanin coloration of the coleoptile, which had numerous missing values, and plant seasonal type and hairiness of external surface of lower glumes (100% monomorphic and 99.7% monomorphic, respectively) improved phenotype dispersion on the biplot, with the first two dimensions explaining over 14% of the variance (Fig. 5).

Plant length contributed most to the first two dimensions (Fig. 6), with additional contributions from ear-related traits such as awn distribution, awn length relative to ear length, and ear glaucosity for the first dimension, and straw pith thickness, glume shoulder shape and width, and glume beak curvature for the second. Traits most distant from the biplot origin, such as plant length and ear traits, were the most variable and provided the highest differentiation among genotypes. Positive relationships, indicated by the alignment of long vectors, were observed between plant length and ear length, ear density, glume beak curvature, and recurved flag leaves. Negative relationships were observed between plant height, on the one hand, and awn length, flag leaf glaucosity, and ear glaucosity, on the other (Fig. 7).

**Table 1   UPOV characteristics, corresponding categories (grades), and genotype frequency distribution in the durum wheat collection.**

| UPOV Characteristic/Grade | 1 | 3 | 5 | 7 | 9 |
|---|---|---|---|---|---|
| CAC—Coleoptile: anthocyanin coloration | absent or very weak | weak | medium | strong | very strong |
| Genotypes (%) | 15 | 0 | 32 | 53 | 0 |
| PGH- Plant: growth habit | erect | semi erect | intermidiate | semi prostrate | prostrate |
| Genotypes (%) | 77 | 23 | 0 | 0 | 0 |
| PRFL—Frequency of plants with recurved flag leaves | absent or very low | low | medium | high | very high |
| Genotypes (%) | 34 | 44 | 17 | 5 | 0 |
| TEE—Time of ear emergence | | early | medium | late | |
| Genotypes (%) | | 49 | 47 | 4 | |
| FLACA—Flag leaf: anthocyanin coloration of auricles | absent or very weak | weak (2) | medium (3) | strong (4) | very strong (5) |
| Genotypes (%) | 27 | 59 | 12 | 2 | 0 |
| FLGLB—Flag leaf: glaucosity of lower side of leaf blade | absent or very weak | weak | medium | strong | |
| Genotypes (%) | 13 | 76 | 11 | 0 | |
| EG—Ear: glaucosity | absent or very weak | weak | medium | strong | |
| Genotypes (%) | 1 | 59 | 40 | 0 | |
| PL—Plant: length | very short | short | medium | long | |
| Genotypes (%) | 0 | 20 | 56 | 24 | |
| EDA—Ear: distribution of awns | awneless | tip awned (2) | half awned (3) | fully awned (4) | |
| Genotypes (%) | 0 | 0 | 72 | 28 | |
| ELA—Ear: length of awns at tip relative to length of ear | shorter | equal (2) | longer (3) | | |
| Genotypes (%) | 1 | 91 | 8 | | |
| LGSS—Lower glume: shape of shoulder | sloping | rounded (2) | straight (3) | elevated (4) | elevated with 2nd beak (5) |
| Genotypes (%) | 2 | 39 | 31 | 26 | 2 |
| LGWS—Lower glume: width of shoulder | very narrow | narrow | medium | broad | |
| Genotypes (%) | 7 | 41 | 35 | 17 | |
| LGLB—Lower glume: length of beak | very short | short | medium | long | |
| Genotypes (%) | 9 | 22 | 38 | 31 | |
| LGCB—Lower glume: curvature of beak | absent | weak | moderate | strong | |
| Genotypes (%) | 62 | 29 | 5 | 4 | |
| LGHES—Lower glume: hairiness of external surface | absent | | | | present |
| Genotypes (%) | 100 | | | | 0 |
| SPCS—Straw: pith in cross section | | thin | medium | thick | |
| Genotypes (%) | | 1 | 99 | 0 | |
| EL—Ear: length (excluding awns) | | short | medium | long | |
| Genotypes (%) | | 0 | 58 | 42 | |
| ED—Ear: density | | lax | medium | dense | |
| Genotypes (%) | | 96 | 4 | 0 | |
| PST—Plant: seasonal type | winter type | alternative type (2) | spring type (3) | | |
| Genotypes (%) | 0 | 100 | 0 | | |

MCA effectively differentiated accessions by plant length into three distinct groups: medium (green), long (blue), and short-stemmed (red) wheats (Fig. 8). Similar grouping patterns, consisting of two partly overlapping groups, were observed for ear glaucosity (Fig. 9) and awn distribution (Fig. 10). Most of the fully awned wheat accessions grouped

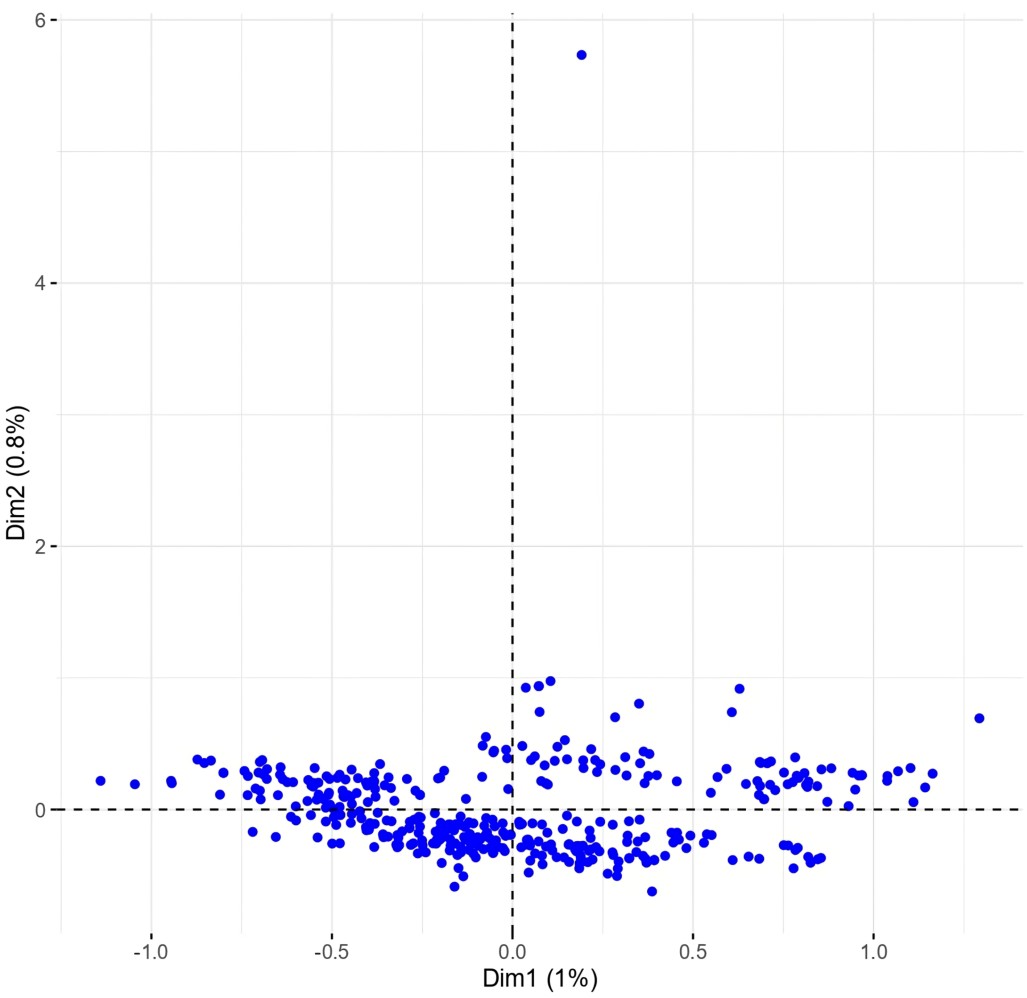

**Figure 4** **Multiple correspondence analysis (MCA) of 370 Western Balkan durum wheat genotypes based on 18 polymorphic morphological descriptors.** Dim1, dimension 1; and Dim2, dimension 2.

with those exhibiting medium ear glaucosity, while the other group included accessions with shorter awns and a less glaucous appearance of ears. Awn length (Fig. 11) revealed a distinct smaller group of genotypes with awns longer than the ear (blue), contrasting with genotypes having equal (green) or shorter (red) awns.

## DISCUSSION

The genetic variability within the durum wheat populations reflected in an average of five phenotypes per landrace (accession) is a well-known evolutionary stress-buffering mechanism that enables stable yields and tolerance to biotic and abiotic stress factors (*Lopes et al., 2015*; *Nadeem et al., 2021*; *Fiore et al., 2019*; *Adhikari et al., 2022*). The geographic clustering in Clusters 2 and 3 suggests the influence of microenvironmental factors on morphological differentiation. Such geographic structuring parallels findings from other studies, such as those in Sicilian durum wheat, where high variability in traits was attributed

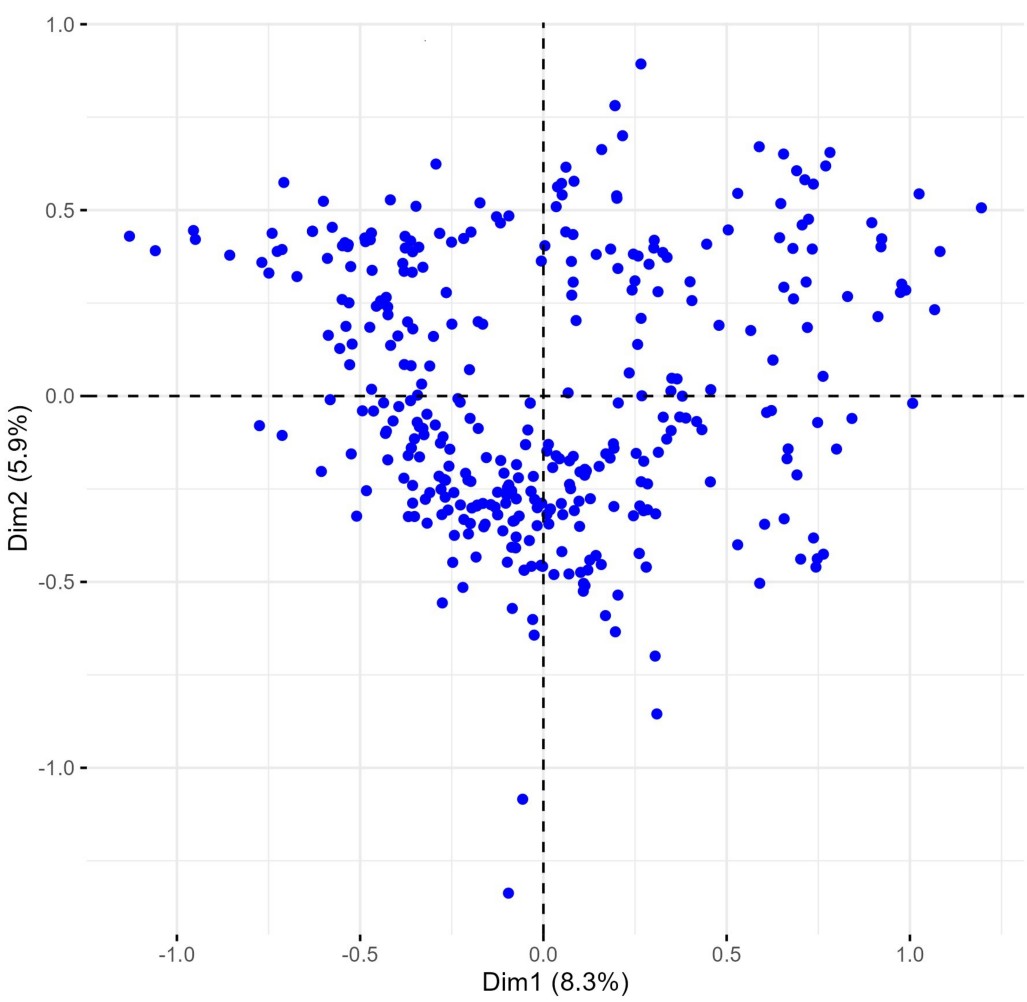

**Figure 5** **Multiple correspondence analysis (MCA) of 370 Western Balkan durum wheat genotypes (without the genotype METD-52/01) with 16 polymorphic morphological descriptors.** Dim1 = dimension 1, and Dim2 = dimension 2.

primarily to genotypic diversity (*Fiore et al., 2019*). Similarly, Ethiopian landrace studies emphasize the role of both genetic and environmental factors in shaping diversity and adaptation (*Mengistu, Kiros & Pè, 2015*). Minimal diversity observed for traits with one predominate category, such as alternative seasonal type, absence of hairiness on the lower glume surface, equal length of awns at tip and the length of ear and lax ear density (with H' values below 0.30) likely reflects adaptive advantages or historical selection pressures, either for these specific traits or for other agronomically important characteristics genetically linked to them. Diversity level is generally considered high when H' > 0.60, intermediate when 0.40 < H' < 0.60 or low when 0.10 < H' < 0.40 (*Eticha et al., 2005*). The low Shannon-Weaver diversity index for these traits suggests that they may represent fixed features within the population, possibly maintained due to their role in conferring stability, stress tolerance, or compatibility with traditional cultivation practices. Lax ear density, for

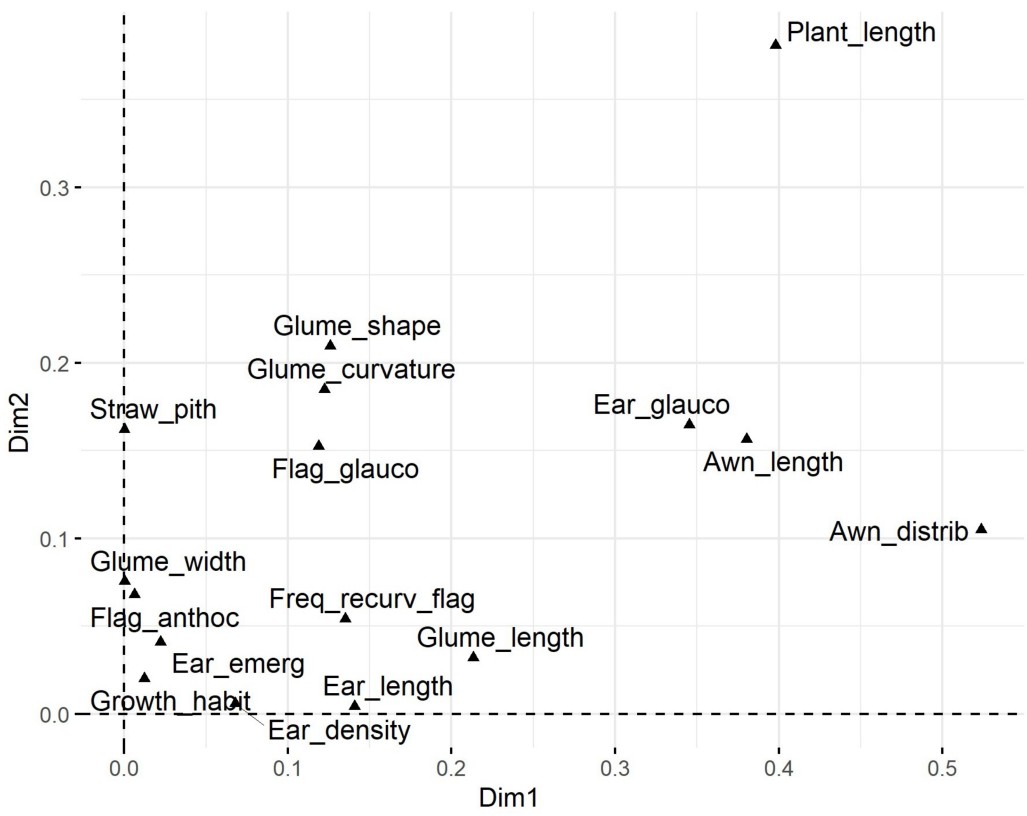

**Figure 6** **Loading plot of 16 polymorphic morphological descriptors based on multiple correspondence analysis (MCA) using the homals R package.** Glume_curvature, curvature of lower glume beak; Glume_length, length of lower glume beak; Glume_width, width of lower glume shoulder; Glume_shape, shape of lower glume shoulder; Awn_distrib, distribution of awns on ear; Ear length, ear length (excluding awns); Awn_length, length of awns at tip relative to length of ear; Ear density, ear density; Ear glauco, ear glaucosity; Ear_emerg, time of ear emergence; Freq_recurv_flag, frequency of plants with recurved flag leaves; Flag_glauco, glaucosity of lower side of flag leaf blade; Flag_anthoc, anthocyanin coloration of flag leaf auricles; Straw_pith, pith in straw cross section; Plant_length, plant length; Growth_habit, plant growth habit. Dim1 and Dim2 represent dimension one and two, respectively.

instance, is associated with open flowering, enhancing cross-pollination and seed dispersal, potentially linked to resilience in heterogeneous environments (*Bayles et al., 2009*). Equal awn length—prevalent in 91% of genotypes—has agronomic significance, contributing to photosynthetic efficiency, grain yield, and drought tolerance (*Maydup et al., 2014*). The average H' value across all traits was 0.62., similarly to that reported for Serbian durum wheat genotypes (H' = 0.616, *Takač et al., 2019*) and slightly lower than Tunisian wheat collections (H' = 0.67, *Ouaja, Bahri & Aouini, 2021*). This overall value masks important variation among trait groups: traits related to flag leaves and glumes showed the highest diversity (H' = 0.74 and 0.66, respectively), while ear- and plant-level traits exhibited more uniform expression (H' = 0.57 and 0.50, respectively), possibly reflecting stabilizing selection or common adaptation strategies within the local agro-ecological context.

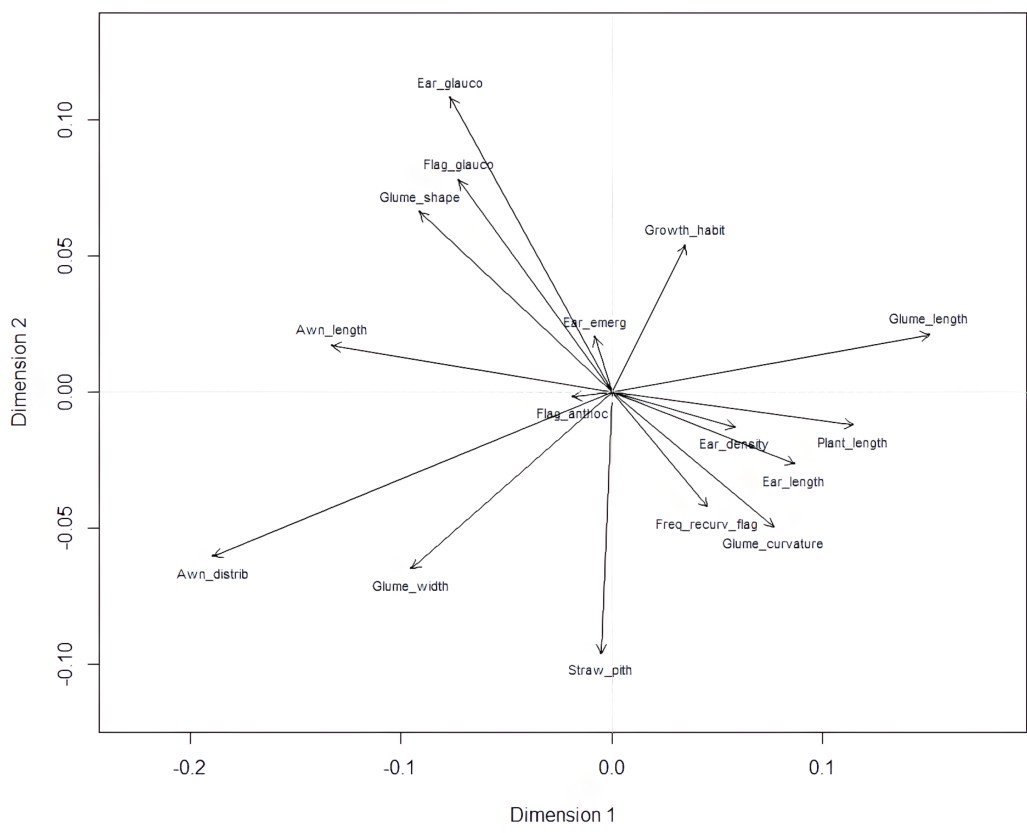

**Figure 7** **Biplot of morphological categories derived from multiple correspondence analysis (MCA) using the FactoMineR R package.** Glume_curvature, curvature of lower glume beak; Glume_length, length of lower glume beak; Glume_width, width of lower glume shoulder; Glume_shape, shape of lower glume shoulder; Awn_distrib, distribution of awns on ear; Ear length, ear length (excluding awns); Awn_length, length of awns at tip relative to length of ear; Ear density, ear density; Ear glauco, ear glaucosity; Ear_emerg, time of ear emergence; Freq_recurv_flag, frequency of plants with recurved flag leaves; Flag_glauco, glaucosity of lower side of flag leaf blade; Flag_anthoc, anthocyanin coloration of flag leaf auricles; Straw_pith, pith in straw cross section; Plant_length, plant length; Growth_habit, plant growth habit. Dim1 and Dim2 represent dimension one and two, respectively.

The low proportion of total variability explained indicates that the categorical variables used in the analysis may not exhibit strong patterns of association, contrasting with other MCA studies that successfully identified distinct clusters or groups within datasets (*Beebe et al., 2001*; *Habtie, Dejen & Dessalegn, 2017*; *Koffi et al., 2021*). This may also be due to the fact that the majority of the analyzed phenotypes belong to the Rogosija lineage and therefore share a similar origin and characteristics. Although our study relied on accessions maintained ex situ, it is important to acknowledge the growing consensus that the long-term conservation and adaptive evolution of landraces are best supported through *in situ* cultivation by traditional farmers. Unfortunately, durum wheat cultivation in Montenegro ceased in the early 1970s, and local landraces have since only been preserved in the national gene bank, with periodic regeneration efforts to maintain seed viability. The absence of farmer-maintained populations limited our ability to include *in situ* samples in

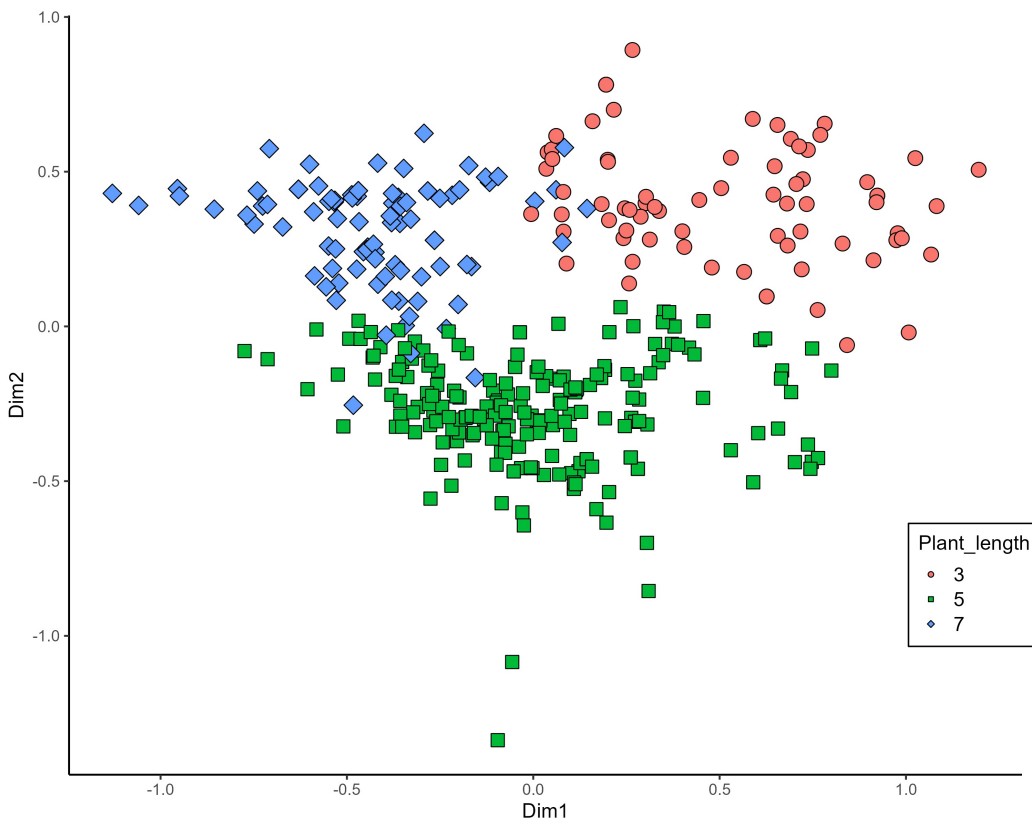

**Figure 8** **Multiple correspondence analysis (MCA) of 370 Western Balkan durum wheat genotypes based on 16 polymorphic morphological descriptors.** Grouping of genotypes for plant length. Dim1, dimension 1; and Dim2, dimension 2.

this analysis. However, we agree that future comparative studies integrating both *ex situ* and *in situ* maintained landraces would provide valuable insights into ongoing evolutionary dynamics, farmer-led selection, and conservation priorities. Given that no traditional farmers currently maintain local durum wheat populations, the gene bank remains the only viable source of this unique genetic material.

Plant height is an important trait closely associated with yield. It is measured from the ground to the top of the canopy at maturity. However, due to stem arching in the field or lodging, which may not always be apparent, this measurement can slightly differ from plant length, which refers to the distance from the base of the stem to the top of the ear on a straightened plant. While plant height has practical agronomic relevance, plant length is a more biological term and is preferred by UPOV as a more precise descriptor. These findings are consistent with previous research in traditional wheat landraces. For instance, *DeLacy, Skovmand & Huerta (2000)* reported strong phenotypic associations between yield components and morphological traits (*e.g.*, spike size, grain weight, maturity) in Mexican landraces, using pattern analysis of unreplicated trials. These co-expressed traits may reflect coordinated selection pressures or adaptation patterns in traditional agroecosystems. Certain morphological traits with their short vectors on the biplot (Fig. 7),

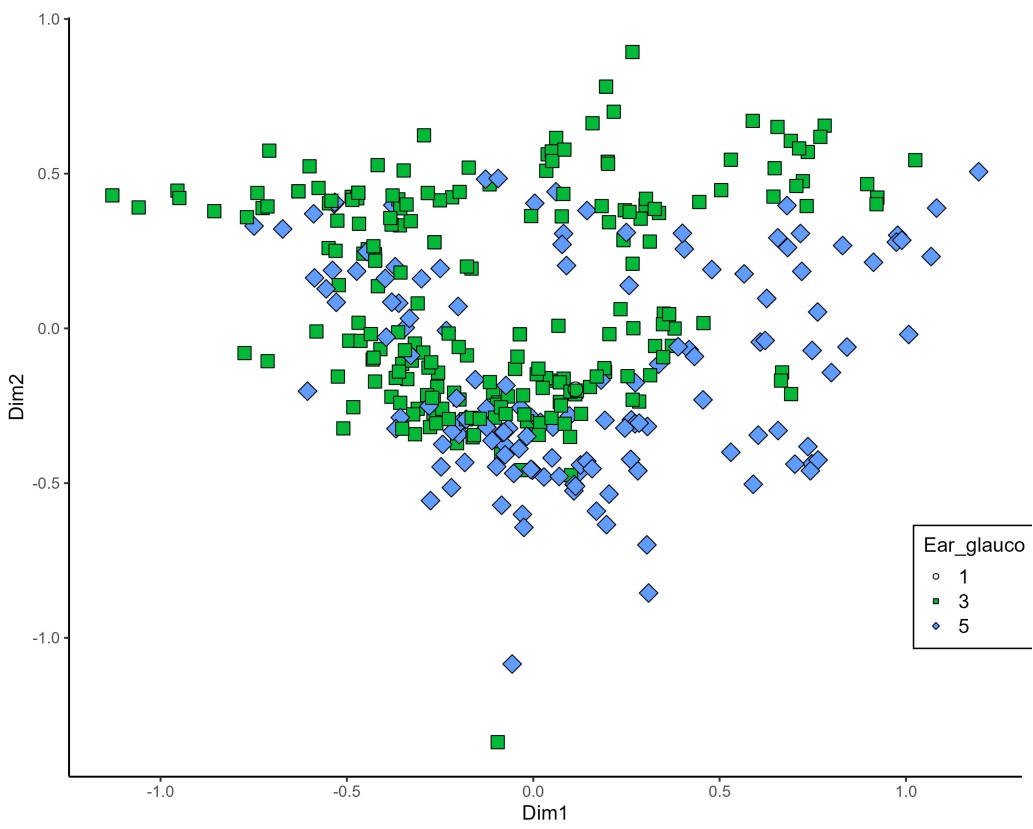

**Figure 9  Multiple correspondence analysis (MCA) of 370 Western Balkan durum wheat genotypes based on 16 polymorphic morphological descriptors.** Grouping of genotypes for ear glaucosity. Dim1, dimension 1; and Dim2, dimension 2.

such as anthocyanin coloration of flag leaf auricles and time of ear emergence, did not contribute importantly to the observed clustering and appear to be unlinked or weakly associated with plant length-related traits, suggesting distinct genetic mechanisms or the influence of different environmental factors. It appears that shorter plants in this durum wheat collection tended to have longer awns and a more intense glaucous layer on their surfaces. Negative correlations imply trade-offs where improvements in one trait may come at the expense of others (*Dwivedi, Reynolds & Ortiz, 2021*; *Liu et al., 2018*). However, these characteristics may work synergistically, as reduced plant height generally lowers lodging risk, while enhanced epicuticular leaf wax deposition has been shown to improve water retention and confer greater drought tolerance in wheat (*Shepherd & Wynne Griffiths, 2006*; *Guo et al., 2016*). Multiple correspondence analysis allowed for the analysis of the pattern of relationships among several categorical dependent variables. Morphological traits in wheat, such as plant height, ear glaucosity, and awn characteristics, play a key role in influencing agronomic performance, environmental adaptability, and yield potential. Plant height has been a major target in wheat breeding programs, particularly during the Green Revolution, when the incorporation of dwarfing genes such as Rht-B1 and Rht-D1 resulted in semi-dwarf cultivars with reduced lodging susceptibility and improved harvest

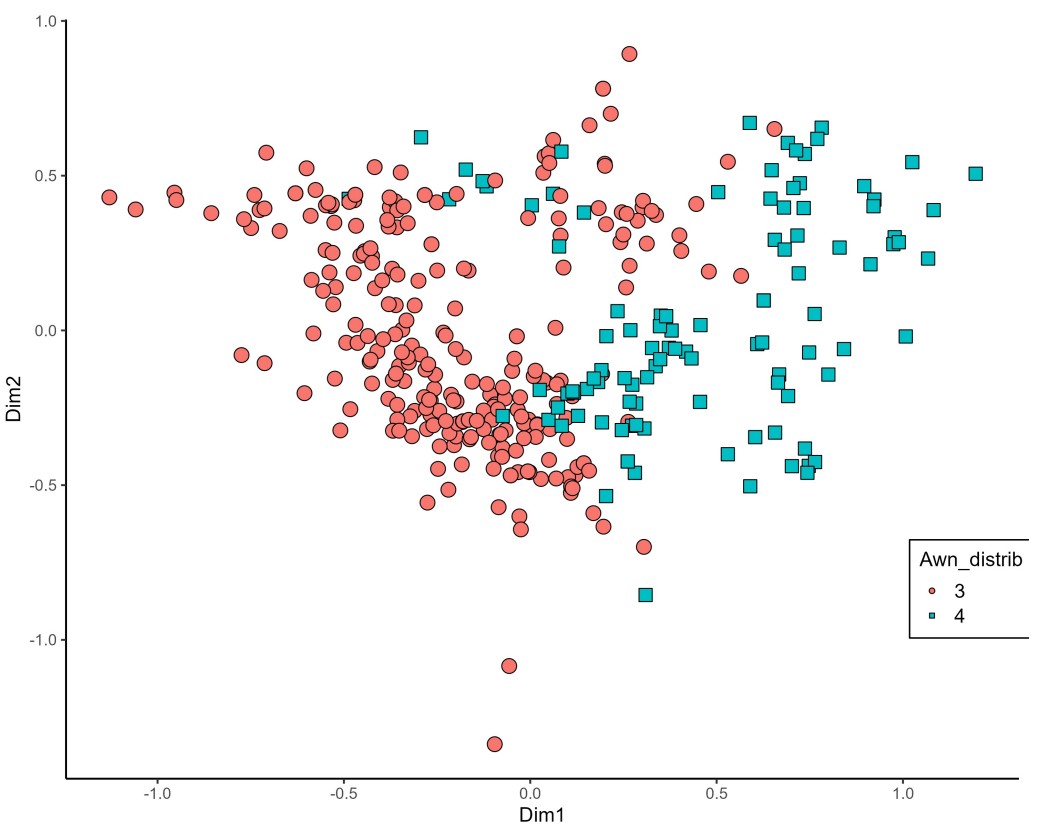

**Figure 10** **Multiple correspondence analysis (MCA) of 370 Western Balkan durum wheat genotypes based on 16 polymorphic morphological descriptors.** Grouping of genotypes for distribution of awns on ear. Dim1, dimension 1; and Dim2, dimension 2.

index by enhancing the allocation of assimilates to reproductive structures (*Ukozehasi, Ober & Griffiths, 2022*). Ear glaucosity, resulting from a waxy epicuticular layer, is associated with improved water-use efficiency and greater tolerance to heat and drought stresses by minimizing transpirational water loss and increasing reflectance of solar radiation (*Zhang et al., 2019*). Awns, which contribute to photosynthetic activity in the wheat ear, are particularly important under stress conditions, as their presence, length, and distribution can enhance grain filling and overall yield (*Rebetzke, Bonnett & Reynolds, 2016*). While morphological traits offer valuable insights into phenotypic variability and adaptive potential, their effectiveness in providing deeper genetic relationships is constrained by environment and limited discriminatory capacity. Nonetheless, recent high-resolution morphological studies on durum wheat landraces confirm the continued relevance of trait-based diversity analyses for conservation and pre-breeding, particularly when integrated with genetic data (*Marzario et al., 2023*). In our study, morphological assessment represents a component of a broader research framework that complements the SNP-based study on a subset of the same durum wheat accessions (*Velimirović et al., 2023*) that identified genetic clusters largely corresponding to the morphological groupings observed in the present analysis. This study builds upon our previous work (*Velimirović*

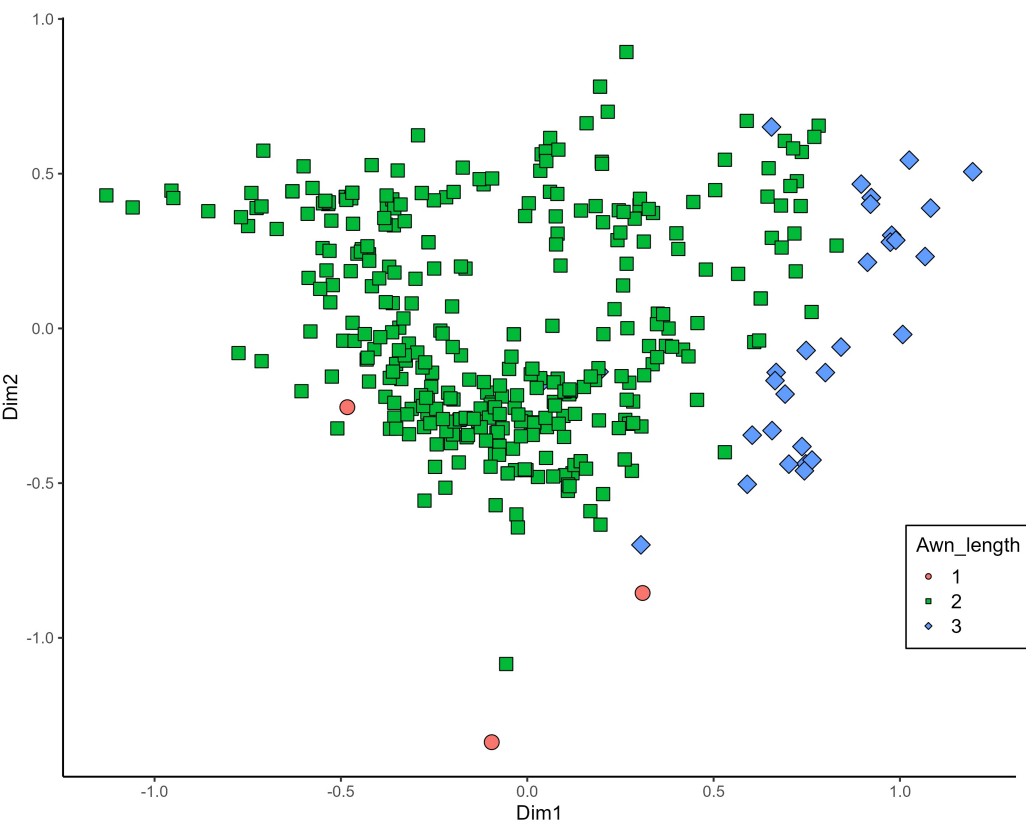

**Figure 11 Multiple correspondence analysis (MCA) of 370 Western Balkan durum wheat genotypes based on 16 polymorphic morphological descriptors.** Grouping of genotypes by trait length of awns at tip relative to length of ear. Dim1, dimension 1, and Dim2, dimension 2.

*et al., 2023*) by expanding both the scope and in-depth morphological characterization. While the earlier study focused on SNP-based genetic structure and included a limited set of binary morphological traits for population clustering within the 'Rogosija' subset, the present analysis covers the full Western Balkan durum wheat landrace collection (80 accessions) and identifies 370 distinct phenotypes using 17 detailed UPOV descriptors. These traits were grouped by plant organs and evaluated using hierarchical clustering, normalized Shannon-Weaver indices, and multiple correspondence analysis. The broader sampling and trait resolution enable finer detection of phenotypic variability and adaptive differentiation, particularly in accessions from Herzegovina, inland Montenegro, and coastal Croatia. Whereas the previous study focuses on genetic structure, the current work emphasizes trait-specific diversity patterns, phenotypic correlations, and morphological features shaped by agro-ecological factors. Together, the two studies offer complementary insights into the structure and utility of local durum wheat diversity, reinforcing the value of integrating genotypic and phenotypic data for conservation and pre-breeding strategies. The concordance between molecular and morphological findings adds value to an integrated approach for robust characterization of genetic resources and for informed pre-breeding strategies. The inclusion of key morphological descriptors in breeding programs remains

vital for the development of durum wheat cultivars with enhanced resilience to stress and sustainable performance under changing climatic conditions.

## CONCLUSIONS

Our study on 80 durum wheat accessions revealed significant genetic diversity and potential adaptive strategies. The identification of 370 differentiated phenotypes highlights the extensive variability within the populations. Key observations, such as the absence of hairiness on the lower glume surface and limited variation in traits like straw pith thickness and ear density, suggest fixed genetic traits and potential selective breeding history. Our findings may point to shared developmental pathway and complex trait interactions in wheat morphology as suggested in previous studies. Understanding these genetic networks is essential for improving wheat through the integration of landraces in breeding programs.

These findings emphasize the importance of considering both morphological diversity and geographical origin when evaluating and utilizing durum wheat landraces. This approach can guide localized breeding programs and conservation strategies aimed at preserving crop genetic diversity and promoting climate-resilient agriculture. Based on these insights, we recommend the development of region-specific seed networks and participatory breeding efforts involving local farmers, which would ensure the sustainable use of genetically diverse, well-adapted landraces in traditional and low-input agricultural systems.

## ACKNOWLEDGEMENTS

The authors wish to express their sincere gratitude to colleagues who contributed to the SNP-based characterization of the Rogosija durum wheat collection, including Dragan Perović and Heike Lehnert (Julius Kühn Institute, Germany), Giacomo Mangini and Mariella Matilde Finetti-Sialer (IBBR-CNR, Italy). Their collaborative efforts in genotyping and preliminary data analysis provided a valuable foundation for the present study.

### Funding
The authors received no funding for this work.

### Competing Interests
The authors declare there are no competing interests.

### Author Contributions
- Ana Velimirović conceived and designed the experiments, performed the experiments, analyzed the data, prepared figures and/or tables, and approved the final draft.
- Sanja Mikić conceived and designed the experiments, analyzed the data, prepared figures and/or tables, and approved the final draft.
- Zoran Jovović conceived and designed the experiments, authored or reviewed drafts of the article, and approved the final draft.

- Dragan Mandić conceived and designed the experiments, performed the experiments, prepared figures and/or tables, contributed tools, and approved the final draft.
- Novo Pržulj conceived and designed the experiments, authored or reviewed drafts of the article, and approved the final draft.

## Data Availability

The raw measurements are available in the Supplementary File.

## Supplemental Information

Supplemental information for this article can be found online at http://dx.doi.org/10.7717/peerj.20068#supplemental-information.

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
