# Peer review of "Morphological diversity of Western Balkan durum wheat landraces"

_PeerJ, doi:10.7717/peerj.20068_

## Round 0.1 · original submission · Major Revisions

Dear Dr. Velimirovic, The manuscript contains many significant shortcomings that were noted by the reviewers. I ask you to correct the shortcomings pointed out by the reviewers very carefully and I hope that this will make it possible to approve a new version of this article.

**Language Note:** The review process has identified that the English language must be improved. PeerJ can provide language editing services - please contact us at [email protected] for pricing (be sure to provide your manuscript number and title). Alternatively, you should make your own arrangements to improve the language quality and provide details in your response letter. – PeerJ Staff

Reviewer 1 ·

Basic reporting

A very good choice of topic, with clearly set goals and hypotheses. In addition, properly selected and cited literature related to the research area.

Experimental design

Well-chosen material with a clear and adequate methodology. Results are properly presented, and the discussion is guided by proper comparison and interpretation of other authors' results.

Validity of the findings

Clearly drawn conclusions based on previously mentioned facts are presented in the results and discussion chapter.

Reviewer 2 ·

Basic reporting

The research addresses an important agricultural issue and provides valuable insights into the diversity of Montenegrin durum wheat. It emphasizes the importance of considering morphological traits and geographical origins in crop diversity studies.

Suggestion:
Conduct a thorough English language edit of the manuscript to correct grammatical errors and improve clarity. Please find some suggestions in the attached file. Summarize redundant sentences and focus on highlighting the critical aspects of the study.

Abstract
It is generally well-written and sums up the study's aims. However, it could be improved by including specific numerical results for better impact and clarity.
Include relevant keywords not presented in the title and abstract to improve searchability.

Introduction
It should be improved, highlighting a solid background on the challenges associated with phenotypic diversity. The knowledge gap and the novelty of the study should be emphasized.

State the novelty of the research, i.e., what differentiates this study from previous research on Montenegrin durum wheat landraces.
Add references to recent studies that highlight the current trends or contrasting results in the field.

The hypothesis and objectives need to be clearly defined.

Experimental design

Methodology
This section appropriately describes the experimental setup, data collection, and statistical analysis. However, it could benefit from visual aids like diagrams or schematics of the experimental design to better visualize and understand.
More information on soil chemical and physical properties, as well as soil particle distribution percentages of clay, silt, and sandy, electrical conductivity, soluble cations, soluble anions, and available nutrients, should be clarified.

Validity of the findings

Results
The section requires significant revision and enhancement. It should begin with a thorough analysis of variance (ANOVA) to establish a foundational understanding of data variability and statistical significance. This will provide readers with a clear context for the subsequent findings.

Discussion
The section would benefit from a more critical analysis of the study's limitations. The manuscript briefly explores these limitations but does not thoroughly explore how they might influence the findings or the broader implications of the research.

Conclusion
This could be improved by reinforcing the study's broader contributions to agricultural practices, particularly in how these findings can enhance the importance of considering both morphological traits and geographical origins in studying crop diversity and adaptation strategies. Based on the study findings, suggest potential policy implications or practical guidelines for farmers.
Follow the journal style guide or citation requirements for uniformity and accuracy. Revise the journal abbreviations in the references for consistency.

Annotated reviews are not available for download in order to protect the identity of reviewers who chose to remain anonymous.

Reviewer 3 ·

Basic reporting

The document has several types and hyphenations to be corrected.
Please specify the annual rainfall at the experimental site and soil type.
Please specify genotypes/accessions used.

Experimental design

The Experimental design used, if there is one, is unclear.
No replications are mentioned.

Validity of the findings

Since the experimental design is unclear, the results are also not properly supported.

In addition, for example, Figure 1 seems to make no sense.

Figure 2 needs to include the names of genotypes.

Text mentioned up to Figure 6, but 12 figures are reported.

Reviewer 4 ·

Basic reporting

Researchers have analyzed morphological variations in durum wheat landrace accessions in Montenegro using morphological markers.

*In the title of the manuscript, plant material is referred to as “Western Balkan durum Wheat Landraces”. But in the manuscript it is stated that 80 Montenegrin durum wheat landraces. Does Western Balkans mean Montenegro? If not, the title is not fit for the content of the manuscript.

Morphological traits can vary greatly among individuals of the same species or even within the same populations. Because morphological traits are greatly affected by environmental factors due to the plasticity of plants. In addition, due to the limited number of different variants in morphological markers, their use in variation analysis is no longer a preferred method.

Experimental design

In the study, seeds obtained from a single spike of each durum wheat accession were manually sown at a depth of 5 cm, in rows of 1 m length and at a distance of 0.2 m between rows, and 80 plots were obtained by sowing seeds from 20 spikes for each accession. In morphological analyses, only 3 plants grown from seeds sown in 20 rows in a plot for each accession were randomly selected and scored. However, this number of samples is not sufficient to perform variation analysis in morphological variation studies. As I mentioned before, since morphological traits are greatly affected by the environment, the number of samples should be greater to reduce the error rate. On the other hand, the number of samples for statistical analyses should not be less than 8 in experimental studies, as this causes error (Cristofolini and Testoni, 2000).
Ref: Cristofolini L. and Testoni M. (2000). The importance of sample size and statistical power in experimental research. A comparative study. Acta of Bioengineering and Biomechanics. Vol. 2 No.1.

”The gene pool that wheat landraces have as genetic resources is an invaluable resource. Preserving genetic resources in gene banks is an important practice. It is important to refresh the seeds stored ex-situ in gene banks at certain intervals and to protect healthy seeds. However, it is very important to preserve wild and landraces of species, especially in cereal plants such as wheat, in situ. Because the evolutionary processes of the seeds in gene banks slow down, their ability to adapt to current external ecological conditions (biotic and abiotic stress factors) also weakens. On the other hand, the fact that wild species continue to grow in natural environments means that their evolutionary processes continue actively. Landraces should continue to be produced in the field by traditional farmers, and in this way, their evolutionary processes continue actively while the genetic diversity in their gene pools is preserved at high levels. In this manuscript, the researchers took the samples from the gene bank and analyzed them. It is important to analyze the seeds in the gene bank, but the genetic diversity they represent will probably depend on the conditions under which they were collected. If the seeds that continue to be produced by traditional farmers were included in the analyses in addition to the seeds taken from the gene bank, the comparative results would be more meaningful."

Validity of the findings

Another important issue is that using only morphological markers in genetic diversity analyses does not provide enough power to discriminate accessions, genotypes, and species, and so on; molecular markers must support these results. In fact, currently even molecular markers such as RAPD and ISSR are not accepted when used alone. In the conclusion and discussion sections of the manuscript, it is claimed that morphological variations provide important results to be used for breeding programs.

“Strong correlations among certain traits suggested potential pleiotropic traits, while others appeared to be governed by distinct genetic mechanisms or environmental factors.”

Pleiotropic traits display complicated heritability. It can not be claimed based on the morphological analyses.

“These findings give valuable insights into the Montenegrin durum wheat diversity, emphasizing the importance of considering both morphological traits and geographical origins in crop diversity studies. Overall, our study provides a foundation for future breeding efforts aimed at enhancing the agronomic performance and resilience of durum wheat cultivars.”

However, in my opinion, this is a very ambitious statement. Because, in addition to morphological markers, more detailed analyses are made with molecular markers and their data are interpreted by taking into account ecological and geographical factors, and parents are selected in breeding programs.
The analyses and evaluations made by the researchers in this study are very valuable. As a morphological analysis study, the experimental design, statistical evaluations, and comments are all appropriate. However, I think that when the number of samples is increased and molecular analyses with molecular markers are added, the article will contribute to the accurate analysis of genetic diversity in durum wheat landraces and to the increase of publication quality.

Additional comments

Minor comments:
“biodiversity” refers to biological diversity, I think it is used instead of genetic diversity. Both terms are used to refer to different levels of diversity. Correct terms should be used.

In lines 68-69: “… permanent loss of 36% of the collection.” A reference should be given for this knowledge.

Annotated reviews are not available for download in order to protect the identity of reviewers who chose to remain anonymous.

---

## Round 0.2 · Major Revisions

Dear Dr. Velimirovic, I ask you to carefully correct the shortcomings pointed out by the reviewers. I hope that the new version of this article will be approved by the reviewers for publication.

·

Basic reporting

The manuscript's introductory section provides a thorough and well-structured overview dedicated to the history, significance, and current state of wheat genetic diversity, with a particular emphasis on local forms of durum wheat in the Western Balkans region. The material presented is informative, timely, and delivered in an engaging manner. The issue of agrobiodiversity loss is of both theoretical and practical importance, rendering the article potentially appealing to a broad scientific audience, including breeders, agronomists, biologists, and genetic resource conservation specialists.
Line 62: The term 'erosion of crop genetic diversity' is used repeatedly in the text without a clear definition or appropriate citation. Despite its extensive utilisation within the extant literature, the incorporation of a concise elaboration or citation to a foundational source (e.g., the Food and Agriculture Organization, biodiversity frameworks, or seminal academic works) would serve to enhance the conceptual clarity and fortify the manuscript, a consideration that is particularly salient for interdisciplinary readerships.
Line 95: The species name T. turgidum should be italicized.

The hypothesis, although formulated in a manner consistent with stylistic conventions, does not fully align with the tenets of scientific logic. Rather than being presented as a tentative assumption to be tested through the study, it is presented more as a statement requiring confirmation. In order to ensure the clarity and coherence of the text, it would be advisable to either transform this fragment into an introductory preamble leading to a clearly articulated research objective, or to explicitly state the hypothesis as a testable assumption the authors intend to examine.
The statement of the research objective necessitates terminological clarification. Phrases such as "descriptor efficiency" and "conservation potential" remain undefined. It is recommended that the criteria used to assess efficiency be specified, and that the term 'conservation potential' be clarified. This is to ensure that there is a common understanding of what is meant by 'conservation potential' — whether it refers to genetic richness, variability level, adaptive value, or other specific attributes.

Experimental design

Line 122: The manuscript states that accessions were collected from the full spectrum of "local agro-ecological zones", yet these zones are not described nor classified. In order to facilitate a more comprehensive understanding of the growing conditions of the samples collected, it is recommended that a concise characterisation of the zones in question be provided, or alternatively, the criteria employed to define them be specified (e.g. climatic, edaphic, altitudinal, etc.).
The experimental design employed in this study involved the use of pseudoreplication, wherein each landrace was represented by a single plot, thus precluding the implementation of true replication. While this may limit the statistical robustness of the phenotypic comparisons and could potentially affect the generalisability of the results, which in some contexts could be grounds for rejecting the manuscript, the authors explicitly acknowledge this limitation by stating: It is acknowledged that the exploratory nature of the study, in conjunction with the limited quantity of seed available, rendered the implementation of a replicated block design or standard control samples impracticable. Therefore, while this concern must be regarded as significant, it may be considered acceptable within the scope of the present exploratory study. Nevertheless, it is strongly recommended that future research includes appropriate experimental replication to enhance the validity and reliability of phenotypic evaluations.
The subsection title "Morphological description" is comprehensible, but it may not fully reflect the analytical and standardized nature of the methodology employed. A more precise term such as "Morphological characterization" or "Assessment of morphological traits" would better correspond to the scientific content presented.
The application of the Shannon and normalized Shannon indices (H, H') to morphological trait data requires further clarification and justification. The development of these indices within the framework of information theory (Shannon & Weaver, 1949) was originally intended to quantify uncertainty or entropy in information systems. Despite their extensive utilisation within the domain of ecology for the purpose of evaluating species diversity based on the relative abundance of discrete entities, the extension of these methodologies to the realm of morphological trait analysis gives rise to a number of significant methodological challenges. It is evident that the dataset under consideration encompasses a combination of both categorical and continuous variables. However, it should be noted that Shannon indices are conventionally employed in the analysis of discrete frequency distributions. The authors should therefore clarify:
• whether all traits were converted into categorical classes prior to analysis,
• how trait frequencies were calculated,
• and provide the precise formulas used for index computation.
Without such information, the interpretation of the index values in the context of morphological variability may be ambiguous. These clarifications are essential for ensuring methodological transparency and reproducibility.

Validity of the findings

The study is based on a substantial volume of empirical data, including 80 durum wheat accessions and the identification of 370 phenotypic variants. The application of appropriate statistical procedures, such as agglomerative hierarchical clustering and multivariate analysis (MCA), has been well-aligned with the research objectives. Moreover, the comparison of the results with existing literature enhances the credibility of the findings. The methodological rigor employed, in conjunction with the evident phenotypic differentiation and geographic traceability of the accessions, serves to substantiate the validity of the conclusions drawn by the authors with regard to the adaptive significance and breeding potential of Montenegrin landraces.

Reviewer 6 ·

Basic reporting

The authors present an extensive morphological characterization of 80 accessions of Western Balkan durum wheat landraces. The study reveals substantial variability across nearly all analysed traits, highlighting the significant potential of these landraces for inclusion in breeding programs. The manuscript is well written, and the introduction provides a solid overview of the current state of knowledge.
The literature on this topic is limited, except for one paper published by the same authors in 2023, “SNP Diversity and Genetic Structure of ‘Rogosija’, an Old Western Balkan Durum Wheat Collection”, in which they performed both morphological and genetic characterization of the same collection used in this manuscript (Velimirovic et al., 2023). The current paper would benefit from a more detailed discussion of its findings to provide better context. Additionally, since a morphological characterization was already carried out in that earlier work, the authors should clearly explain how the present study differs, emphasizing any novel aspects or expanded analyses.

These are some minor reviews:
-Lines 111, 114, 215, and the caption for Supplementary Table 1 refer to “Montenegrin.” Instead, in the title and in all the manuscript “Western Balkan” was used. For consistency, it would be preferable to use “Western Balkan” throughout the manuscript.
-Figures 5, 8, 9, 10, and 11: the captions state that the Multiple Correspondence Analysis (MCA) is based on 17 polymorphic morphological descriptors. However, in lines 220–222, the authors mention that an outlier and the descriptor for anthocyanin coloration of the coleoptile (due to numerous missing values) were excluded, resulting in 16 descriptors.
-Supplementary Table 2: The unit of measurement for precipitation is not reported.
-Supplementary Table 3: In the first column replace “platns” with “plants”

Velimirović, A., Jovović, Z., Perović, D., Lehnert, H., Mikić, S., Mandić, D., Pržulj, N., Mangini, G., & Finetti-Sialer, M. M. (2023). SNP Diversity and Genetic Structure of “Rogosija”, an Old Western Balkan Durum Wheat Collection. Plants, 12(5), 1157. https://doi.org/10.3390/plants12051157

Experimental design

The experimental design is generally well described. The authors appropriately acknowledge some limitations that affect the statistical significance of their results.
There are only some minor points to review.
In line 139, it is stated that “three individuals from each row were selected”; it would be important to clarify whether these individuals were chosen randomly, selected for their similarity, or if any atypical plants were intentionally excluded.
In the manuscript, it is mentioned that “there are scientific studies that have successfully employed UPOV descriptors to characterize wheat during a single growing season in combination with molecular diversity assessments (Marzario et al., 2023).” However, Marzario et al. (2023) used a randomized complete block design in two different locations, which differs significantly from the design used in this study. Therefore, this reference appears inappropriate in this context and should be removed or replaced with a more relevant one.
The version of R used, along with the appropriate reference, should be provided. Additionally, the versions of the Homals and FactoMineR packages, as well as their relevant references, should also be reported.
Lê, S., Josse, J., & Husson, F. (2008). FactoMineR: An R Package for Multivariate Analysis. Journal of Statistical Software, 25(1), 1–18. https://doi.org/10.18637/jss.v025.i01
de Leeuw, J., & Mair, P. (2009). Gifi Methods for Optimal Scaling in R: The Package homals. Journal of Statistical Software, 31(4), 1–21. https://doi.org/10.18637/jss.v031.i04
Marzario, S., Sica, R., Taranto, F., Fania, F., Esposito, S., De Vita, P., Gioia, T., & Logozzo, G. (2023). Phenotypic evolution in durum wheat (Triticum durum Desf.) based on SNPs, morphological traits, UPOV descriptors and kernel-related traits. Frontiers in Plant Science, 14, 1206560. https://doi.org/10.3389/fpls.2023.1206560

Validity of the findings

The results are generally clear and well presented, with one minor point needing clarification: in line 221, the authors mention “numerous missing values” for the trait “anthocyanin coloration of the coleoptile,” but Supplementary Table 1 does not appear to include any missing values for this descriptor. The authors should clearly report the number of missing values and specify whether these were included in the analyses or excluded.
The discussion is well developed and comprehensive, clearly describing the results. However, there is one point that should be further expanded. As mentioned above, this study is closely connected to Velimirovic et al. (2023). While this link is briefly acknowledged in lines 320–325, the discussion section would benefit from a deeper comparison of the results of the two studies, highlighting both the similarities and the differences in findings and their implications.

---

## Round 0.3 · Minor Revisions

Dear Dr. Velimirovic, I ask you to make minor changes recommended by the reviewer before the article is accepted.

·

Basic reporting

The authors have implemented all the recommendations of the reviewer. The quality of the manuscript has been significantly improved. I recommend the article for publication.

Experimental design

The authors have implemented all the recommendations of the reviewer. The quality of the manuscript has been significantly improved. I recommend the article for publication.

Validity of the findings

The authors have implemented all the recommendations of the reviewer. The quality of the manuscript has been significantly improved. I recommend the article for publication.

Reviewer 6 ·

Basic reporting

Authors accepted all the revisions and modified the manuscript accordingly. The effort made by the authors to address the revisions has helped improve the quality of the paper.
Now it is acceptable for publication in PeerJ Life & Environment.

Experimental design

Just one minor point: in lines 232–234, the R and Homals versions are still missing, although the response to the reviewers states that they were added.

Validity of the findings

No comment

---

## Round 0.4 · accepted · Accept

The article can be recommended for publication.

Reviewer 6 ·

Basic reporting

no comment

Experimental design

All of the reviewer’s recommendations have been addressed by the authors

Validity of the findings

no comment